# Human West Nile Meningo-Encephalitis in a Highly Endemic Country: A Complex Epidemiological Analysis on Biotic and Abiotic Risk Factors

**DOI:** 10.3390/ijerph17218250

**Published:** 2020-11-08

**Authors:** Mircea Coroian, Mina Petrić, Adriana Pistol, Anca Sirbu, Cristian Domșa, Andrei Daniel Mihalca

**Affiliations:** 1Department of Parasitology and Parasitic Diseases, University of Agricultural Sciences and Veterinary Medicine of Cluj-Napoca, 400372 Cluj-Napoca, Romania; cristiandomsa@gmail.com (C.D.); amihalca@usamvcluj.ro (A.D.M.); 2Department of Infectious Diseases, University of Medicine and Pharmacy “Iuliu Hațieganu” of Cluj-Napoca, 400012 Cluj-Napoca, Romania; 3Department of Physics, Faculty of Sciences, Ghent University, B-9000 Ghent, Belgium; mpetric@avia-gis.com; 4Department of Physics, Faculty of Sciences, University of Novi Sad, 21102 Novi Sad, Serbia; 5Avia-GIS NV, 2980 Zoersel, Belgium; 6National Centre for Communicable Diseases Surveillance and Control, National Institute of Public Health, 050463 Bucharest, Romania; adriana.pistol@insp.gov.ro (A.P.); anca.sirbu@insp.gov.ro (A.S.); 7Romanian Ornithological Society, Cluj Office, 400336 Cluj-Napoca, Romania

**Keywords:** abiotic, biotic, mosquito, modelling, predictors, West Nile virus, WNND

## Abstract

West Nile virus (WNV) is one of the most prevalent mosquito-borne viruses. Although the infection in humans is mostly asymptomatic, 15–20% of cases show flu-like symptoms with fever. In 1% of infections, humans develop severe nervous symptoms and even die, a condition known as West Nile neuroinvasive disease (WNND). The aim of our study was to analyze the influence of abiotic and biotic factors with the human WNND cases during the period 2015–2019. A database containing all the localities in Romania was developed. Abiotic and biotic predictors were included for each locality: geographic variables, climatic data, and biotic factors. Spatial distribution of the WNND infections was analyzed using directional distribution (DD). The Spearman’s rank correlation coefficient was employed to assess the strength of association between the WNND infections and predictors. A model was generated using the random forest ensemble learning method. A total number of 535 human WNND cases were confirmed in 308 localities. The DD showed a south-eastern geographical distribution. Weak correlation was observed between the number of human WNND cases for each year and the predictors. The highest predicted probability was around urbanized patches in the south and southeast. Increased surveillance and control measures of vectors in risk areas should be implemented and educational campaigns should be made available for the general public in order to raise awareness of the disease and inform the population about prophylactic measures.

## 1. Introduction

West Nile virus (WNV) is one of the most prevalent mosquito-borne viruses, belonging to the *Flaviviridae* family, in the Japanese encephalitis serocomplex. It is included in the genus *Flavivirus*, together with other viruses with public health relevance such as dengue, Zika, Japanese encephalitis, yellow fever, and Usutu [1]. Two genetic lineages are known, lineage 1 and lineage 2, the first with three subclades [2].

The natural life cycle involves birds as reservoir hosts and ornithophilic mosquitos, mainly from the genus *Culex* as vectors [3]. Even though humans and horses can get infected, they are considered dead-end hosts, due to the low and short-term viremia [4]. Although the infection in humans is mostly asymptomatic, 15–20% of the cases show flu-like symptoms with fever, known as the West Nile fever. In 1% of the infections, humans develop meningoencephalitis with severe neurological symptoms and even die, a condition known as West Nile neuroinvasive disease [1,5].

After its description in Uganda [6] and subsequent reports in Africa [7,8], the first reports in Europe were in France [9] and Russia [10], followed by many others. Currently, the virus is present in most of the continent [11]. In Romania, the first report of WNV infection was in 1996, as a major human epidemic, with 393 cases, of which 352 had acute meningoencephalitis [12]. The virus was detected again in 1997 and 1998 [13], but with a limited number of infections. The next significant outbreak was in 2010 with 50 confirmed cases and a fatality rate of 10% [14]. Viral circulation and human infections were recorded also in the following years [15], with a third significant outbreak in 2016, with 93 neuroinvasive cases [16]. One of the most significant outbreaks in Europe (2083 confirmed cases), including Romania (277 cases and 43 deaths), was recorded in 2018 [17]. Finally, 66 infections were reported in Romania in 2019 [18].

Several studies attempted to identify risk factors associated with WNV infection in humans [2,19,20]. A wide variety of biotic and abiotic factors play an important role in maintaining the vector-pathogen-host cycle. Abiotic factors influence the vector abundance but also their vectorial competence [21]. Higher temperatures directly influence the viral circulation, by increasing the mosquito populations, the frequency of feeding and the oviposition rate. Moreover, higher temperatures control the pathogen’s development rate inside the mosquito, leading to a shorter extrinsic incubation period [2]. Several studies have highlighted the role of the water bodies in the vector density, serving as breeding sites, especially in early summer [5]. The unclear role of precipitation is given by the different environmental conditions required, depending on the mosquito species. On the one hand, precipitations will favor the presence of stagnant water, leading to an increase in the vector density, by providing favorable conditions for the larval stages, but heavy rain conditions can remove the larvae [21]. In the case of *Culex pipiens,* drought conditions may favor its development due to its preferences for eutrophic, stagnant water, which is richer in nutrients. The role of drought is also discussed from the perspective that it can lead to increased interactions between vectors and hosts, due to diminished water sources [21].

Among biotic factors, avian host-related factors, such as distribution and abundance, are crucial for viral circulation, their ability to maintain a high and prolonged viremia being the reason why they are named amplifying hosts [22]. It is assumed that migratory birds are responsible for introducing the virus from Africa in Europe, and also for its dispersal, but local and resident bird populations are also needed for viral amplification [19,23].

Most studies evaluating the influence of abiotic factors included only short-term seasonal data, with only one study from the USA analyzing average multi-seasonal climatic factors [24]. Herein, we propose such an approach using long term averages. Additionally, despite being a highly endemic country, no attempts have been made so far to evaluate the risk factors associated with WNV outbreaks in Romania. Romania is a country with diverse ecological conditions, comprising five ecoregions [25], a high diversity of vector mosquitoes [26] and a significant number of bird species [27]. Romania is also a major stopover for migratory birds [28]. Moreover, to the best of our knowledge, there is no comprehensive analysis on the possible influence of non-human related biotic factors on the epidemiology of human West Nile neuroinvasive disease (WNND) infections in Europe. In this context, the aim of our study was to analyze the integrated influence of various multi-annual abiotic and biotic factors with the human WNND cases over a period of five years (2015–2019).

## 2. Materials and Methods

### 2.1. Data Source

A database containing all the lowest level administrative units in Romania (known as communes) (*n* = 3186) was developed using data from the official website of the Romanian National Agency of Cadaster and Real Estate Advertising [29]. For the purpose of this study, each of the six administrative units (sectors) of the capital city, Bucharest, were considered a separate locality. The total surface of each assessed locality was calculated according to the official limits using a Geographical Information System GIS approach; the polygons’ area of administrative units was calculated in Stereo 70 national projection. The following abiotic and biotic predictors were included for each locality: (1) geographic variables: average altitude (m), surface of flowing water bodies (ha), surface of still water bodies (ha); (2) climatic data: annual average humidity (%), average spring temperature (March–May) (°C), average summer temperature (June–August) (°C), maximum average summer temperature (°C), average annual precipitations (mm/m²); biotic factors: human population, horse population, domestic fowl population, and estimated abundance of four wild bird species: house sparrow (*Passer domesticus*), Eurasian collared dove (*Streptopelia decaocto*), western jackdaw (*Coloeus monedula*), and hooded crow (*Corvus cornix*).

Data regarding altitude was derived from the free digital elevation model datasets, made available by WorldClim [30], based on the SRTM (Shuttle Radar Topography Mission’s digital elevation models/3D) datasets. The surface of flowing water bodies and the surface of still water bodies were calculated using the Corine Land Cover datasets, provided by the Copernicus Land Monitoring Service [31]. The climatic data used for the statistical analysis was downloaded from the WorldClim database, which hosts high spatial resolution global weather and climate data. The data used are long term averages (1970–2000) and are standard climatic (e.g., maximum temperature) or derived bioclimatic variables (e.g., average spring months temperature). In addition, we imported another variable, relative humidity (annual mean moisture index), provided by Climond database [32], a derived dataset based on the general bio-climatic data (climatic data combined for ecological purpose in species distribution).

The number of human inhabitants per locality was extracted from a spatial database supplied by the Romanian National Institute of Statistics ([33]—Computer System of the Register of Territorial Units). The data on horse and domestic fowl population were provided by the regional Veterinary Authorities.

The bird species abundance data used in this study are the results of distribution modelling of the targeted species. Bird data were provided by the Common Bird Monitoring programme, implemented by the Romanian Ornithological Society and Milvus Group. These were combined with environmental data provided by the WorldClim database and Copernicus Land Monitoring Service and cover both habitat and climatic predictors. Based on monitoring data and ecological variables for each species, several models were fitted in order to assess the best species distribution. Species distribution modelling were performed in R programming language, using random forest niche modelling. Mathematical averaging of the obtained values was performed to gain a unique individual value for each locality (spatial unit). The resulting mean values were included in the statistical analysis (unpublished data).

The number of annual human cases of WNND per locality from 2015 to 2019 was provided by The National Centre for Surveillance and Control of Communicable Diseases Bucharest.

To obtain the annual cumulative presence for 2015–2019 we analyzed the total number of human WNND cases per locality and year, and the annual positivity rate per locality—positive (1)/negative (0). If one or more human cases were recorded in a locality, it was considered positive in that year, each locality receiving a multiannual score from 0 to 5, depending on the number of years in the period 2015–2019 in which it was positive.

### 2.2. Statistical Analysis

#### 2.2.1. Correlation

The Spearman’s rank correlation coefficient was employed to assess the strength of association between the annual number of human WNND cases and the predictor variables following the standardized criteria suggested by Evans [34]. This score is invariant to scaling of either variable or shifts in the mean, thus the pre- or post-processing of the variables does not change the nature of their association. Moreover, the score is dimensionless and scaling with regards to measuring units does not affect the correlation value.

Both Pearson’s product-moment correlation coefficient and the Spearman’s rank correlation coefficient were considered, and the Spearman’s score was chosen as the more suitable test statistic for this analysis of not strictly linear association. This nonparametric score is more resilient to outliers and considers both linear and non-linear monotonic relationships between the input variables [35].

#### 2.2.2. Directional Distribution

The spatial distribution of the human WNND annual presence was analyzed using the directional distribution (DD) tool in QGIS (Open Source Geospatial Foundation (OSGeo), Zürich, Switzerland) [36]. The DD ellipse is a graphical analysis tool that allows for the summary and measurement of directional trends, dispersion, and central tendency of WNND presences for each year. One DD ellipse covers approximately 68% of the presences in the annual datasets.

#### 2.2.3. Random Forest

The WNND presence probability model was generated using the random forest (RF) ensemble learning method [37]. RF is a versatile data mining technique that can model dynamic relationships between various predictors with one of the most accurate classifiers [37,38]. RF can be used for classification and regression as well as to assess variable importance. The RF model was built in R-project software (Bell Laboratories, Murray Hill, Union County, NJ, USA) [39]. The input dataset was balanced prior to training. Class imbalance can heavily influence both the model training and the validation process with failure in classification often occurring with skewed training sets [40,41]. Artificial balance samples were generated using a smoothed bootstrap approach following the methodology proposed by Menardi and Torelli [40]. This balancing method represents a trade-off between underand oversampling balancing approaches. It retains the observed conditional densities in the presence and absence classes, thus reducing the risk of overfitting and increasing model generalizability, which is often compromised by oversampling methods. The Gini impurity criterion was used to assess variable significance, whereby the lower the Gini impurity number, the higher the node classification accuracy. The Gini number is a measure of the likelihood of an incorrect classification of a randomly chosen element if it was randomly classified according to the observed distribution of the class labels from the dataset. Random subsets of predictor variables were considered during the training at each split, and the variable which had the lowest Gini impurity number was chosen. To indicate the significance of each variable the mean decrease in the Gini impurity index was calculated. All available data were fed to the RF during training, and the estimated WNND probability therefore depicts the overall classification of each pixel as suitable or unsuitable for the occurrence of WNND based on the cumulative agreement of the training predictors. Prediction was done over a regular grid with 30-arc s resolution.

#### 2.2.4. Model Validation

The Kappa statistic and the sensitivity and specificity metrics are the most common scores for the validation and verification of machine learning models. Cohen’s kappa coefficient represents how well the instances classified by the RF classification algorithm correspond to the observed data, controlling for the baseline accuracy of a random classifier. A score >0.81 is considered excellent, 0.61–0.80 as substantial agreement, 0.41–0.60 as moderate agreement, 0.21–0.40 fair agreement, 0.00–0.20 slight, and <0 as poor agreement [42].

The relationship between sensitivity, the fraction of true positives (number of true positives/(number of true positives + number of false negatives)), specificity, and the fraction of false negatives (number of true negatives/(number of true negatives + number of false positives)) indicates how well the model is performing for the presence and absence classes.

The out-of-bag (OOB) error, or misclassification rate, is employed to calculate the internal prediction error of the RF model. The verification of the RF model was performed on an independent validation set. From the original data, 70% was used to train the model and 30% for testing and validation, preserving the relative ratios of the presence and absence classes.

## 3. Results

During 2015–2019, a total number of 535 human WNND cases were confirmed in Romania. We excluded from the database two cases from 2016, from Bucharest, due to the lack of the specific location (sector). Overall, in the interval of our analysis, 308 localities were reported with at least one positive case (Table 1). Out of these, during 2015–2019, in 267 localities WNND was reported only once, in 26 twice, in 7 localities three times, four times in 5 and five times in 3. Both the number of human WNND cases and the number of positive localities showed the same trend, with the lowest values in 2015 and the highest in 2018 (Figure 1). The number of new localities that were negative the previous year and the number of localities that turned negative the following year are shown in Table 1.

The directional distribution was generated for each year individually between 2015 and 2019. The standard deviational ellipse showed a south-eastern geographical distribution, most of the human WNND cases being located in this region (Figure 2). The same trend was observed when compiling the entire time period between 2015 and 2019 and analyzing the annual cumulative presence (*n* = 1, 2, 3, 4, or 5 years) for each locality (Figure 3).

A weak correlation was observed between the number of human WNND cases for each year during 2015–2019 and the biotic and abiotic predictors (Spearman’s rank correlation coefficient) (Figure 4, Appendix A). Higher correlations were obtained when comparing the number of cases for the entire period (2015–2019) with the predictors (Figure 5). The average altitude and the annual average humidity showed negative weak correlations. The average temperature in spring, average temperature in summer, Eurasian collared dove, and western jackdaw abundance showed weak positive correlations. We found the same weak correlations when comparing the annual cumulative presence (0–5) with the predictors (Appendix A).

A moderate positive correlation (0.3 < r < 0.8, *p* < 2.2 × 10^−16^) was recorded between the number of years with human WNND cases (0–5) and the number of human cases for each year, with the lowest correlation for 2015 and the highest correlation for 2018 (Table 2).

The correlation of cumulative presences (0–5) (Figure 6) and the predictors displays a similar pattern to the correlation for the total number of human infections for 2015–2019 and the predictors (Figure 5), as shown in Appendix A (Appendix A).

In order to have a better understanding of the correlations that could affect the statistics, biotic and abiotic predictors were correlated with each other (Figure 7). A relationship was observed between temperature and relative humidity with altitude, reflecting their vertical profile (negative gradient for temperature, positive gradient for humidity). In addition, an expected correlation was found between the different temperature derivatives (average spring temperature, average summer temperature, maximum average summer temperature). The Eurasian collared dove abundancy had a strong positive association with western jackdaw abundancy; a moderate positive correlation was detected between the rest of the wild bird abundancy models with each other.

The predicted probabilities of human WNND occurrence in Romania are shown in Figure 8. The higher probability area is located in the south and southeast part of the country. The lowest probability for human WNND occurrence is located in the high-elevation region along the Western, Eastern, and Southern Carpathians. The Transylvanian plateau has a slightly higher predicted probability of human WNND presence. Several patches of higher predicted probability are present along the western border. The highest predicted probability is around urbanized patches in the southeast, especially in the highly populated area around Bucharest.

Based on the mean decrease Gini impurity index, the human population per locality was identified as the most important variable for predicting human WNND presence in Romania (Figure 9). Precipitation and altitude scored in the second and third position, respectively. Abiotic factors regarding temperature and relative humidity ranked between four and eight. The western jackdaw probability was identified as the most important biotic factor apart from human population.

The performance of the RF model was evaluated using an independent validation set; the main verification metrics are given in Table 3. The balanced accuracy or percent correctly classified is 94%. A Cohen’s index of 0.906 can be classified as almost perfect agreement with the validation set according to the benchmark categories defined by Altman et al. [43]. The sensitivity and specificity indicate that there is good discrimination between the presence and absence classes. The internal out-of-bag error rate was 4.57% for 500 trees and a sub-set of 3 variables at each split.

## 4. Discussion

In this study we aimed to identify the biotic and abiotic predictors able to influence the annual and total number of human WNND cases in an endemic country [16], but also to understand the determinants that influence the annual positivity in given areas.

The number of human WNND cases and the number of positive localities showed expected trend similarities, due to the small number of infection cases per locality per year. This led to similarities between the cumulative presences and the predictors, respectively, and the total number of human cases and the predictors, showing small differences in correlations. The moderate positive correlation, between the cumulative presence and the number of cases for each year could suggest the endemic circulation of the virus in some localities.

Temperature has a positive effect on mosquito populations, by influencing the reproduction, feeding, and the breeding season. Moreover, the risk of transmission of WNV is increased by higher temperatures, due to the shortening of the incubation period [2]. Paz et al. [19] highlighted the importance of the temperature positive anomalies in a West Nile disease outbreak, when positive correlations were found in several European countries, including Romania. In the light of the weak correlations we obtained between general (e.g., maximum average summer temperature) and combined (e.g., average spring temperature) long-term temperature data, our assumption is that the human WNND outbreaks could be influenced by the temperature anomalies in particular years.

Another factor that can influence outbreaks is the viral circulation in Africa [13,23], as well the overwintering rate of the mosquitoes in the winter preceding the transmission season [44]. Our weak correlation regarding the relative humidity and precipitation strengthen the inconsistent role of these abiotic factors in the epidemiology of WNV [2,19].

WNV has an endemic circulation in the south and south-eastern part of Romania, where significant outbreaks were registered in the past [12,15,16]. From one epidemic to another, there have been a large number of cases with severe forms of illness and an increased death rate. According to our analysis, most of the human cases were located in this part of the country, between 2015 and 2019.

The mean decrease Gini index identified the human population as the most important predictor and the random forest regression model showed that the highest probability of an outbreak occurs near highly urbanized areas. This is evident in the south-eastern part of the country, especially in Bucharest, where certain sectors have recorded human cases for up to five consecutive years throughout the study period. Previous studies have also identified high incidence rates in Bucharest [45]. This was also proposed by Bradly et al. [46], who demonstrated a higher prevalence of anti-WNV antibodies among songbirds when the degree of urbanization increased.

*Culex* spp. ornithophilic mosquitoes are considered to be some of the most important vectors for WNV transmission [47]. According to Savage et al. [45], *Culex pipiens* was previously identified as the main vector for WNV in Romania. A study conducted in the United States demonstrated that *C. pipiens* feeds mainly on birds, followed by humans [48]. Highly urbanized areas are characterized by low biodiversity but sometimes high densities [49]. Hence, the vertebrate hosts for mosquitoes are limited to a few species, unlike in rural areas where the host spectrum is more diverse. This has been demonstrated also for other vector-borne pathogens such as *Borrelia* spp., *Anaplasma phagocytophilum*, or *Rickettsia* spp., which show a remarkably high prevalence in ticks collected from urban environments [50].

The role of birds in viral maintenance and amplification has been already established, and several studies were previously conducted in Romania showing different levels of seropositivity among wild and domestic birds [51,52]. Additionally, the important avian migration routes that pass-through Romania were incriminated [45]. The birds were used as predictors due their important role as amplifying hosts [53], their long and high viremia, as well as their proximity to humans.

To the best of our knowledge, this is the first study that compares the human WNND cases with the wild bird species abundance. Our analysis included four bird species, all with a demonstrated role in the epidemiology of WNV. Apart from the human population, the second most important biotic factor was represented by the western jackdaw abundancy. Lim et al. [54] showed that jackdaws are susceptible to both WNV lineages (1 and 2) that are currently circulating in Europe. Other corvids are also important reservoirs and amplifying hosts for the WNV [55], especially in the New World [56,57]. House sparrows are distributed worldwide and very common in urban areas, with their density being positively influenced by the degree of urbanization [58]. Their role as important reservoir host was previously established by Komar [59] in a study conducted in the US. Previous studies conducted in Egypt showed that both house sparrows and hooded crows are able to develop high viremia [60]. Eurasian collared doves are globally distributed, and experimental infections conducted in the US showed that they may develop viremia levels capable of infecting mosquitoes [61]. Despite the proven role that birds play in WNV epidemiology, our results have not been conclusive.

In this study we highlighted the south-eastern distribution of the human WNND cases in Romania, over a period of five years (2015–2019), and our statistical model using random forest regression showed the same evolution trend especially around highly urbanized areas. We registered weak correlations between human WNND cases and the biotic and abiotic predictors, but using the mean decrease Gini index the human population, western jackdaw abundancy, total precipitations, and altitude ranked first. Additionally, this is the first comprehensive study in Romania regarding the abiotic and biotic factors that could influence the distribution of WNND infections among humans.

## 5. Conclusions

Increased surveillance and control measures of vectors should be implemented for better disease management. Educational campaigns should be made available for the general public in order to raise awareness of the disease and inform the population of prophylactic measures. Romania does not benefit from an integrated surveillance system for WNV, so possible links between bird abundance and human infections could provide valuable information. Awareness campaigns, targeted surveillance and control of mosquitoes should be considered by the authorities mainly, at least in the identified risk areas.

## Figures and Tables

**Figure 1 ijerph-17-08250-f001:**
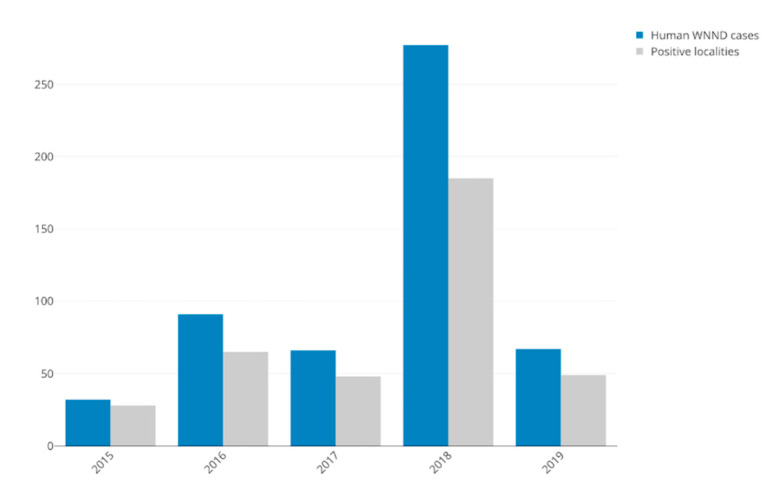
Annual number of human West Nile neuroinvasive disease (WNND) cases and number of positive localities.

**Figure 2 ijerph-17-08250-f002:**
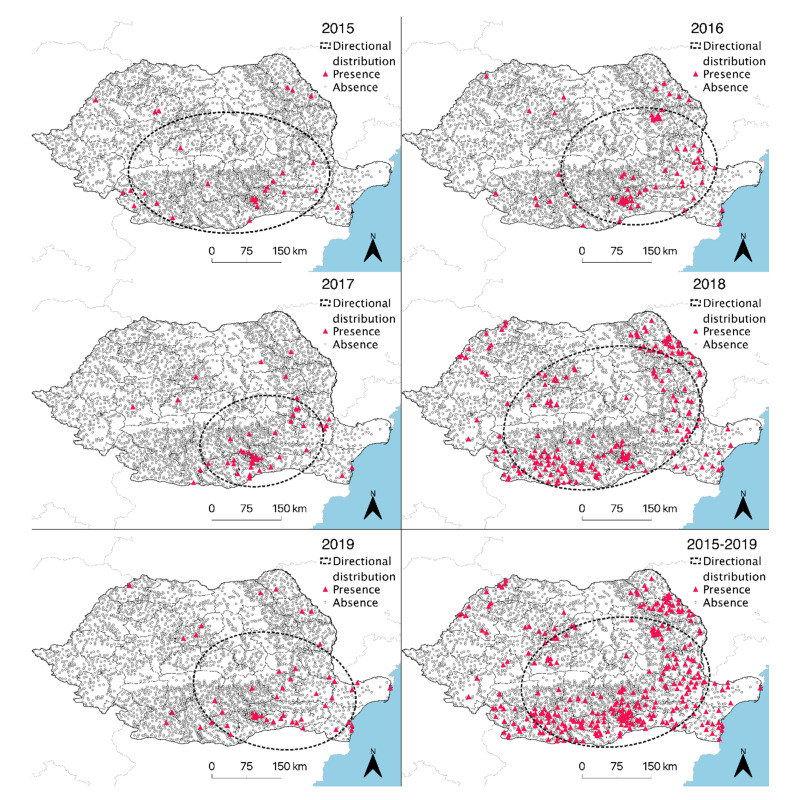
Directional distribution of human WNND cases in Romania between 2015 and 2019.

**Figure 3 ijerph-17-08250-f003:**
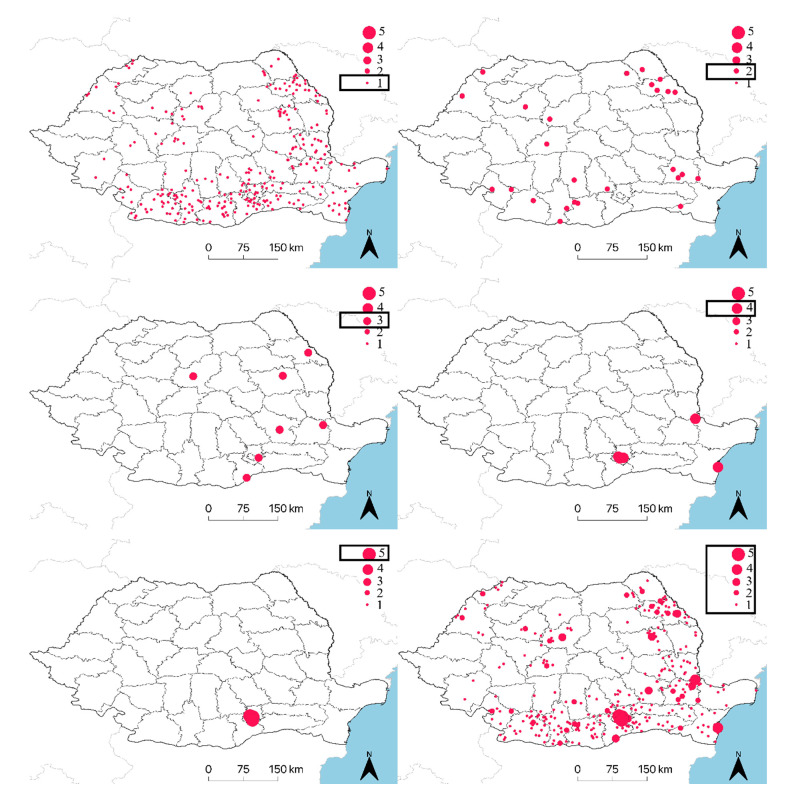
Annual positivity of human WNND cases for year 1, 2, 3, 4, and 5 and cumulated cases, between 2015 and 2019.

**Figure 4 ijerph-17-08250-f004:**
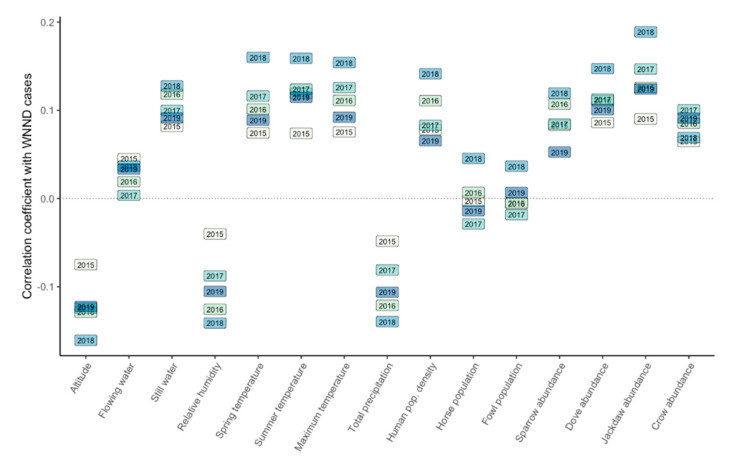
Spearman’s rank correlation coefficient between the number of human WNND cases and the studied predictors for each year.

**Figure 5 ijerph-17-08250-f005:**
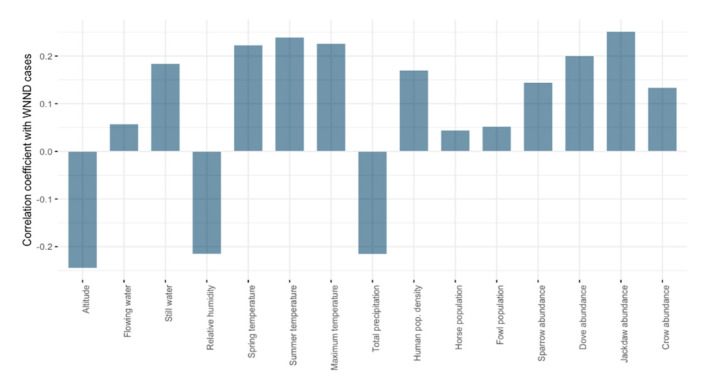
Spearman’s rank correlation coefficient between the number of human WNND cases and the studied predictors for the entire period (2015–2019).

**Figure 6 ijerph-17-08250-f006:**
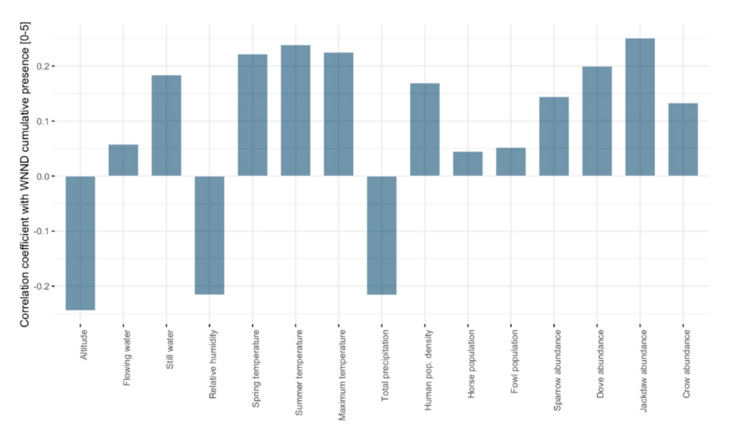
Spearman’s rank correlation coefficient between the cumulative presence 2015–2019 [0–5] and the predictors.

**Figure 7 ijerph-17-08250-f007:**
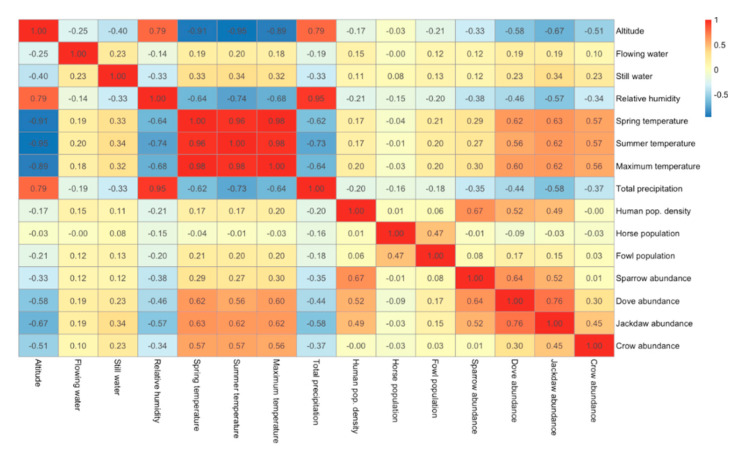
Spearman’s rank correlation coefficient between the biotic and abiotic predictors.

**Figure 8 ijerph-17-08250-f008:**
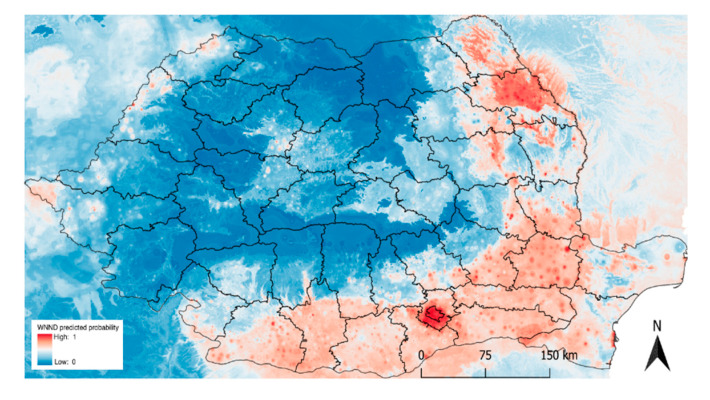
Predicted probability of human WNND presence obtained from RF ensemble learning model.

**Figure 9 ijerph-17-08250-f009:**
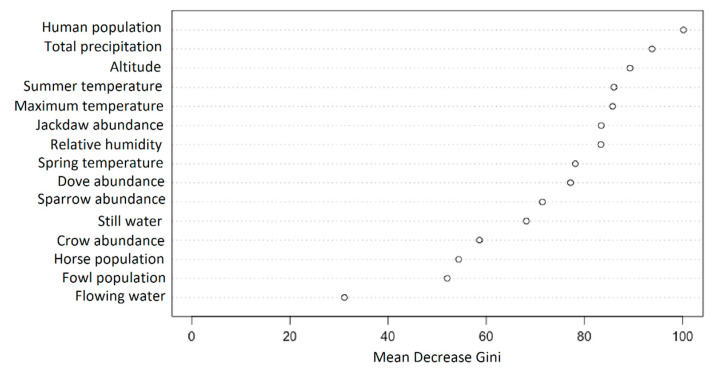
Variable importance for predicting the presence of human WNND cases.

**Table 1 ijerph-17-08250-t001:** Number of human WNND cases, positive localities, and trend overview.

Variable	2015	2016	2017	2018	2019	Total
Total Human WNND Cases	32	93	66	277	67	535
Total Positive Localities (% of total localities)	28 (0.87%)	65 (2.04%)	48 (1.50%)	185 (5.80%)	49 (1.53%)	308 *
New Positive Localities (percent out of annual total)	not calculated	59 (90.8%)	36 (75%)	170 (91.9%)	33 (67.3%)	-
Localities that Turned Negative in the Next Year (percent out of annual total)	22 (78.6%)	53 (81.5%)	33 (68.75%)	169 (91.4%)	not calculated	-

* localities with cases in more than one year were calculated only once.

**Table 2 ijerph-17-08250-t002:** Spearman’s rank correlation coefficient between the number of years with human WNND cases (0–5) and the number of human cases for each year.

Year	2015	2016	2017	2018	2019
Spearman’s rank correlation coefficient	0.2972029	0.4517637	0.3884571	0.7636387	0.3927736

**Table 3 ijerph-17-08250-t003:** Random forest (RF) validation results.

Variable	Balanced Accuracy	Sensitivity	Specificity	Cohen’s Kappa	McNemar’s Test *p* Value
Verification metrics	0.94	0.911	0.967	0.906	0.0002671

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
