# Peer review of "Human West Nile Meningo-Encephalitis in a Highly Endemic Country: A Complex Epidemiological Analysis on Biotic and Abiotic Risk Factors"

_ijerph, 2020, doi:10.3390/ijerph17218250_

Round 1

Reviewer 1 Report

This is a well-designed and authored study. I have only minor comments, as follows:

Line 48 - I believe you mean "neurological" rather than "nervous."

Line 72 - "do [to] it's preferences..."

Lines 80-81 - I am aware of at least one study of WNV using long-term seasonal data similar to the current study. See Young et al. “A Remote Sensing and GIS-Assisted Landscape Epidemiology Approach to West Nile Virus.” Applied Geography 45 (December 2013): 241–49. https://doi.org/10.1016/j.apgeog.2013.09.022. I would recommend revising this section to remove the claim that yours is the first to use long-term seasonal data.

Line 98 - "projection" rather than "project"?

Lines 123-126 - This appears to be a run-on sentence that should probably be split into two sentences.

Line 154 - "ellipse [is] a graphical..."

Line 155 - Did you mean "WNND" instead of "WNV" here? Please review the usage of both throughout the paper to ensure consistency.

Line 180 - This is incorrect. Specificity is (number of true negatives / (number of true negatives + number of false positives)).

Figure 3 - It is difficult to distinguish the line features on the map. Perhaps the lines outside Romania could be removed or faded, or you could add the national border in a bold line to help those less familiar with the region. A more prominent national border could also be helpful on Figure 8.

Figure 4 - I haven't seen this type of figure shown in quite this way before. I like it - nicely done.

Lines 265-266 - This sentence confused me. Consider revising for clarity.

Line 338 - I'm not sure it's fitting to use the term "evolution" here since that was not really a focus of the study. Perhaps you meant distribution or even infection?

Lines 340... - The Conclusion section feels a bit tangential. Perhaps reframing it o mention how your results could be used to inform the suggested changes, including surveillance or education campaigns.

Reviewer 2 Report

In this manuscript Coroian et al analyzed the influence of abiotic and biotic factors with the human WNND cases during the period 2015–2019. They found weak correlations between human WNND cases and the biotic and abiotic predictors, but using the Mean Decrease Gini index the human population, western jackdaw abundancy, total precipitations and altitude ranked first. Overall, this is the first comprehensive study in Romania regarding the abiotic and biotic factors that could influence the WNV evolution among humans and fairly completed. 

Author Response

Dear reviewer,

Thank you very much for the timely response and for the comments.

Reviewer 3 Report

Generally, the paper is well written and was interesting to read. The methods used are appropriate. I have the following comments and suggestions.

Minor typographical suggestions

line

Sentence

Suggestion

63

aredirectlyinfluencing

directly influence

69

the precipitations 

precipitation

115

therelativehumidity

relativehumidity

154

The DD  ellipse a

The DD  ellipse  is  a

187

cases have been confirmed

cases were  confirmed

197

one year were calculate only once.

one year were calculated only once.

Comments /questions

  1. in line 81 you mention that you propose to “analyze multi-seasonal climatic factors”. Is this true? You will average annual data with long term averages.
  2. In line 87 you say “comprehensive meta-analysis “. This is not really a meta-analysis in classical sense
  3. In the different statistical methods employed, clearly state the data used. Cumulative / annual presence/ absence
  4. in line 163 under the methods section, you mention ‘The input dataset was balanced prior to training’, how was this done. Describe explicitly and reasons.
  5. Define Gini impurity in brief for the readers
  6. In line 169, you write ‘each pixel ’. Was prediction done over a regular grid, if so mention in methods
  7. You mention ‘weak’ and ‘better’ correlations, what does these mean?
  8. In line 237 you say “to exclude biases that could affect the statistics, biotic and abiotic predictors were correlated with each other” This does not exclude the bias, just shows the magnitude. You could check variance inflation and remove highly collinear variables.
  9. Could you consider using population density and not just population to see if the results still hold.
  10. Another way to validate the predictive ability of the model is to reserve one year, for example 2019 and assess how well the model generalizes. Try this and report
  11. In the methods you don’t indicate how yearly trend is captured in the models
  12. Would you assess the nonlinear relationships between some of the covariates such as temp and West Nile occurrences?
  13. I wonder given the data, you can fit a Poisson model as sensitivity analysis.
